# The Influence of Serpentine Soil on the Early Development of a Non-Serpentine African Thistle, *Berkheya radula* (Harv.) De Wild

**DOI:** 10.3390/plants11182360

**Published:** 2022-09-09

**Authors:** C. J. Roebuck, S. J. Siebert, J. M. Berner, J. Marcelo-Silva

**Affiliations:** Unit for Environmental Sciences and Management, North-West University, Potchefstroom 2520, South Africa

**Keywords:** bioaccumulation, development, heavy metals, metallophyte, photosynthetic efficiency

## Abstract

Serpentine soils are rich in heavy metals and poor in nutrients, limiting plant species’ performance and survival. Nevertheless, specificities of such limitations as well as adaptability features required for thriving in serpentine environments are barely known. The Barberton Greenstone Belt in South Africa is an example of an area containing serpentine soil with adapted vegetation. In this study, a pot experiment was performed to compare development features (i.e., germination rates, leaf count, leaf length, biomass and photosynthetic capacity) during the early development of the non-serpentine species *Berkheya radula*, a genus consisting of known metal hyperaccumulators from serpentine areas in South Africa. *B. radula* was grown in serpentine soils taken from the Barberton region. *B. radula* leaves had heavy metals in concentrations that confirmed the species as a phytoextractor. There were trends for enhanced productivity and photosynthesis in the serpentine treatments compared to the control. Leaf count, leaf length, electron transport efficiency (ψ_Eo_/(1 − ψ_Eo_), density of reaction centers and PI_ABS,total_ were significantly and positively correlated with at least one of the heavy metals in the leaves. Germination rates were positively influenced by K, whereas biomass and the density of reaction centers were negatively affected by Ca and P, and only Ca, respectively. The heavy metals Zn, Ni and Co were positively correlated with each other, whereas they were negatively correlated with the macronutrients K, Ca and P. The latter correlated positively with each other, confirming higher fertility of the control soil. Our study suggests that *B. radula* exhibits metallophyte characteristics (i.e., preadapted), despite not naturally occurring on metal-enriched soil, and this provides evidence that the potential for bioaccumulation and phytoremediation is shared between serpentine and non-serpentine species in this genus.

## 1. Introduction

South Africa has regions with ultramafic-derived soil known as ‘serpentine soil’. These soils can be broadly characterised by increased levels of heavy metals and lack of plant nutrients, such as P, N and K [1,2]. The soil is often rich in Ni, Cr and Mg at levels considered toxic to most plant species [1,3,4]. It also has a greater Mg:Ca ratio than other soils, making nutrient uptake problematic for species that are not adapted to metal rich soils [2]. The increased levels of Mg, for instance, increases the competition between this and other elements for receptors on plant roots and inhibits or decreases the ability to effectively take up nutrients. Such soil conditions have promoted habitat specialisation and edaphic endemism within serpentine environments [3,4,5].

The influence of serpentine soil on the development of plants is yet to be fully understood, as there are many discrepancies regarding its effects on the growth and development of serpentine adapted and non-adapted species [5,6,7]. Serpentine soils are characterised by plant species (i.e., metallophytes) adapted to toxic concentrations of heavy metals in the soil, including hyperaccumulators, such as *Berkheya coddii* and *B. nivea* [8]. *Berkheya* is one of only two genera (the other being *Senecio*) in Southern Africa that hyperaccumulate Ni (>1000 ppm in dry leaf tissue), and it might be possible that this trait is phylogenetically shared [9].

Metalliferous soils create extremely harsh environments for plants to grow in; therefore, these plants need to be specifically adapted [7,10]. These soils are characterised by high phytotoxicity, with metallophytes presenting metal tolerance, exclusion or accumulation of heavy metals [11]. A metallophyte is defined as a plant that holds a bioaccumulation factor greater than one [11] and can occur as one of two types, namely obligate or facultative [12].

This study aimed to determine the effects of serpentine soils on the growth and development of a non-serpentine species, *B. radula*, to study their growth capabilities on metal rich soil in comparison to serpentine species currently utilized for phytoremediation. This species is common in alluvial black soils, which are not characterised by high heavy metal content [13,14]. We hypothesised that serpentine soils will negatively affect the growth and vitality of *B. radula* as it is not adapted to metalliferous soil. Understanding the effects of serpentine soils on non-adapted species may improve our knowledge of the effects of metal-rich soils on plant development in stressed environments.

## 2. Results and Discussion

### 2.1. Heavy Metals and Macronutrients

The serpentine soil contained the expected heavy metal and macronutrient levels for this type of ultramafic substrate (Appendix A) [15,16]. With the exception of Zn, heavy metals were all significantly higher in the serpentine soil treatments (Figure 1). Cr, Ni and Mn were the main contributors to the distribution and grouping of the soil samples among treatments (Table 1; Figure 2). These metals usually result from metasomatised and hydrothermally altered ultramafic rocks, from which serpentine soils derive [15,17]. The macronutrient Mg was significantly higher in the serpentine soils, whereas K was higher in the control. The pH was neutral in the control and near neutral in the serpentine soils, as expected [16]. Heavy metal concentrations in *B. radula* leaves differed regarding Ni, Sr and Zn—all higher in plants grown in the serpentine soils. The distribution of the concentrations was similar to that of the soils, with Ni, Mn and Zn as the main elements driving the grouping. Macronutrients Ca, K and P were higher in the control treatment, which reflected their lower availability in serpentine soil.

The similarities found in type and concentrations of heavy metals and macronutrients in soils and leaves were in accordance with previous studies on the uptake of elements by plants in South Africa [11]. Unexpectedly, *B. radula*, a non-serpentine species, was able to take up these elements from the serpentine soils, and even showed a trend of higher productivity and photosynthetic capacity.

Individuals of *B. radula* grown in S1 and S2, with a higher Mg:Ca ratio in the soil, takes up more Mg in the leaves. The reason for this is possibly due to ion exchange between the plant root and the soil particles [17]. Soil particles are negatively charged, which facilitates the binding of Ca and Mg ions [17,18]. These ions compete for the same cation exchange sites on soil particles and ion transporters on the plasma membrane. When there is limited Ca and increased Mg (as with serpentine soil), Mg can outcompete Ca on the uptake sites on the plasma membrane. Plants on serpentine soil deal with low Ca and high Mg either through Mg exclusion (high affinity for Ca transport) or Mg tolerance once in the leaf such is possibly the case here.

In a similar fashion, an increased level of K in the soil will increase the bioavailability of Mn [17], and this could be verified throughout all three treatments. The difference of pH ranges may allow for the easier movement of metals in and out of the soils [19,20], as well as for a high Ca content which may be increasing the availability of other nutrients in the soil.

### 2.2. Early Development and Photosynthetic Capacity

Germination rates, biomass, leaf count and leaf length were similar among treatments (Table 2). Productivity was generally higher on the serpentine treatments, but not significantly (*p* > 0.05; Figure 3). Increases in morphological parameters (e.g., leaf length) have been observed in plants growing in nutrient-deficient soil, as a mechanism to improve the photosynthetic area for instance [21]. Here, however, biomass was negatively correlated (*p* < 0.05) with the macronutrients P and Ca (higher in the control soil), whereas leaf count was positively correlated (*p* < 0.05) with Ni and Co, and leaf length positively correlated with Zn and Co (Table 3). A decrease in biomass was observed in P-sensitive plants under high Ca content [22]; however, the biochemical steps involved are still unknown, and there are no records of such a process in *Berkheya*.

In terms of photosynthetic capacity, plants from the serpentine treatments generally outperformed those from the control treatment. The chlorophyll *a* fluorescence transients of plants of all treatments were similar at 90 d.a.p.; however, they became significantly distinct at 180 d.a.p. (Figure 4). This became particularly evident in the multiple turnover steps, between the I and P inclination points (Figure 4). The PI_total_ performance index was also higher in the serpentine treatments after 90 d.a.p. (Table 2). The PI_total_ and its partial parameters were positively correlated with heavy metals in the leaves (Table 3), whereas the density of the active reaction centres [γ(RC)/(1 − γ(RC))] was negatively correlated with Ca. The influence of Ca on the photosynthetic machinery involves multi-step events specifically related to the control of the pH in the thylakoid lumen, through bindings with calcium sensor proteins [23]. However, how a decrease in the density of reaction centres in *B. radula* could occur remains to be investigated.

The role of heavy metals in plant development cannot be understood without considering substrate concentrations, bioavailability and species-specificities which together determine the uptake, translocation and effects on plant morphology and physiology [15,19]. The three heavy metals Co, Ni and Zn showed positive and significant correlation to leaf length and PI_total_. Co and Ni are indirectly involved in photosynthetic activity, mainly through transpiration influence and N metabolism, respectively [24,25], whereas Zn is an important factor on the chloroplast development and the repair of photodamaged D1 proteins in the photosystem II [26]. At the same time, Co, Ni and Zn are also well known for their toxicity to plants [27,28,29]. The three metals are taken up by Fe transporters [30,31] and their hyperaccumulation and translocation are dependent on Fe concentrations in the plant [32]—in this study, uptake in serpentine soils was increased, although not significantly. Co and Ni concentrations were higher in serpentine soils when compared to that of the control substrate, but *B. radula* was able to cope with and seemingly benefit from elevated concentrations. This is characteristic of metallophytes—plants that can tolerate and make use of metals in enriched edaphic environments [33]. This makes such species ideal for green technologies such as phytoremediation of contaminated soils [34].

*Berkheya* has three known hyperaccumulators in South Africa [14]. It is possible that the results observed for *B. radula* reflected shared metallophyte traits within the genus lineage—not necessarily the hyperaccumulation feature (e.g., bioaccumulation > 1000 ppm Ni), but mostly metal tolerance and good performance in metal-toxic/nutrient-poor soils. This preadaptation has been demonstrated for edaphic specialists that may encompass variation that could serve as the raw material for speciation on atypical, harsh habitats [35]. Serpentine and non-serpentine populations of a species may be constitutively adapted to serpentine [36], and our study of *B. radula* shows that this type of preadaptation may also hold for non-serpentine species of a genus comprising many serpentine-tolerant species.

Another possible explanation would be an occasional inadvertent uptake of heavy metals as a consequence of efficient nutrient scavenging mechanisms [37]. Co, Ni and Zn were positively correlated with each other and negatively correlated with the macronutrients Ca and P (although not always significantly) in the plant leaves (Table 3). This means that they could have all been translocated within the plant by the same process. Inadvertent uptake can be observed in plants growing in nutrient-deficient soil [37], but this would still not clarify why these metals improved the photosynthetic performance of *B. radula*.

## 3. Materials and Methods

### 3.1. Sampling Sites

Two sampling sites were chosen along the Barberton Greenstone Belt in South Africa. The first sampling site (S 25°49.501′ E 30°48.430′), referred to as ‘serpentine 1′, was situated in grassland surrounded by pine plantations and was rich in moist red laterite soil (Figure 5). The second sampling site (S 25°40.021′ E 31°02.584′), referred to as ‘serpentine 2′, and was located in a savanna surrounded by mines on dry, rocky, grey soil. Soil was collected at a depth of 20 cm and 40 kg was collected from each site.

### 3.2. Pot Experiment

Seeds of *B. radula* were collected from non-serpentine areas within the urban perimeter of Potchefstroom (South Africa) and sowed in three different treatments: control, serpentine 1 and serpentine 2 soil. The control treatment was composed of a commercially available gardening mix which comprised a mixture of soil, sand, compost, coir, perlite and vermiculite. Four 4L pots were used per treatment (i.e., replicates) and each pot received 10 seeds (∑n seeds = 10 seeds per pot × 4 pots × 3 treatments = 120 seeds). The pots were arranged in a completely randomised design in a greenhouse. The day and night temperatures were set to 25 °C, under photosynthetic active radiation that ranged between 600 and 800 μmol m^−2^ s^−1^. The plants were watered once a day for 60 s by an irrigation system of sprayers. The seedlings were assessed 90 days after planting (d.a.p.) and again at 180 d.a.p.

### 3.3. Heavy Metal Analysis

Soil samples (∑nsoil = 3 samples × 3 treatments = 9) were dried, ground and analysed for the following heavy metals: Cr, Cu, Fe, Mn, Ni, Sr, Zn, and the macronutrients Ca, K, Mg and P, using a Thermo Scientific Niton XL3t GOLDD+ handheld X-ray fluorescence instrument. Each sample was analysed three times to ensure the accuracy of the measurements. The pH was estimated via 1:2.5 extraction.

Plant leaf samples (∑n leaves = 3 samples × 3 treatments = 9) were washed with deionised water, dried, ground and weighed to approximately 50 mg per sample. This was followed by acid-digestion with HNO_3_ (9 mL, 65%) and HCl (3 mL, 32%), and micro-waved digestion at 200 °C (Milestone, Ethos UP, Maxi 44) for 15 min. Samples were analysed with an ICP-MS (Agilent 7500 series) for the determination of the same elements as the soil.

### 3.4. Early Development and Photosynthetic Capacity

Developmental features were measured as follows: (1) germination rate—proportion of germinated seeds per pot for each treatment (seed germination recorded when hypocotyl breaks though the soil surface); (2) leaf count; (3) leaf length—measured from the base to the apex of the leaves; and (4) plant biomass—plants were harvested and dried in an oven at 60 °C for 72 h and then weighed on a precision scale.

The photosynthetic capacity of the plants was quantified by measuring the efficiency of photosystem II by means of chlorophyll *a* fluorescence. Measurements were taken with a Handy PEA fluorimeter on 1 h dark-adapted leaves. The OJIP transients were analysed using the PEA Plus V1.10 software (Hansatech Instruments Ltd., King’s Lynn, UK). Leaves (∑nleaves = 3 leaves × 3 plants × 3 treatments = 27 leaves) were illuminated by red light (650 nm peak) of 3000 µmol photons m^−2^ s^−1^ and recorded for 1 s with a 12-bit resolution. The performance indicator, PI_ABS,total_, and the four partial parameters it comprises were used to assess the photosynthetic capacity of the plants: the maximum yield of primary photochemistry, [ϕ_P0_/(1 − ϕ_P0_)]; the probability of moving an electron further than the primary quinone acceptor (Q_A_^−^) into the electron transport chain, [ψ_E0_/(1 − ψ_E0_)]; the probability of electron transfer from the intersystem electron carrier to the electron acceptors at the PSI acceptor side, [δ_R0_/(1 − δ_R0_)] and the density of the active reaction centres, [γ_RC_/(1 − γ_RC_)].

### 3.5. Data Analysis

Heavy metals, macronutrients, and developmental and photosynthetic parameters were evaluated individually among the treatment groups by means of one-way ANOVA or Kruskal–Wallis tests, depending on the distribution of the data (Shapiro–Wilks, α = 0.05). Differences among turnover steps and curves on the OJIP tests were evaluated through ANOVAs with repeated measurements (α = 0.05). Tukey tests (for ANOVAs) and multiple comparisons of *p*-values (for Kruskal–Wallis) were applied in the presence of significant differences as a *posthoc* method (α = 0.05). Principal component analyses were also applied to soil and leaf data to identify the main heavy metals contributing to the differences among treatments. Correlations (Pearson, α = 0.05) were investigated among heavy metals and macronutrients in leaves, as well as among these and the developmental and photosynthetic parameters, in the search for possible causations of the observed results.

## 4. Conclusions

*Berkheya radula* is a non-hyperaccumulator; however, this study provides evidence of its ability to uptake, tolerate and beneficially use heavy metals, allowing it to be categorised as a metallophyte. Furthermore, *B. radula* showed trends of better developmental and photosynthetic performance in serpentine soils compared with plants in the control. *B. radula* offers proof of inherent preadaptation to soil rich in heavy metals. It highlights the sharing of traits between serpentine and non-serpentine species within the genus *Berkheya*. Hence, further studies are needed to investigate the developmental characteristics of the genus in metal-contaminated soils to explore its potential for future use in phytoremediation.

## Figures and Tables

**Figure 1 plants-11-02360-f001:**
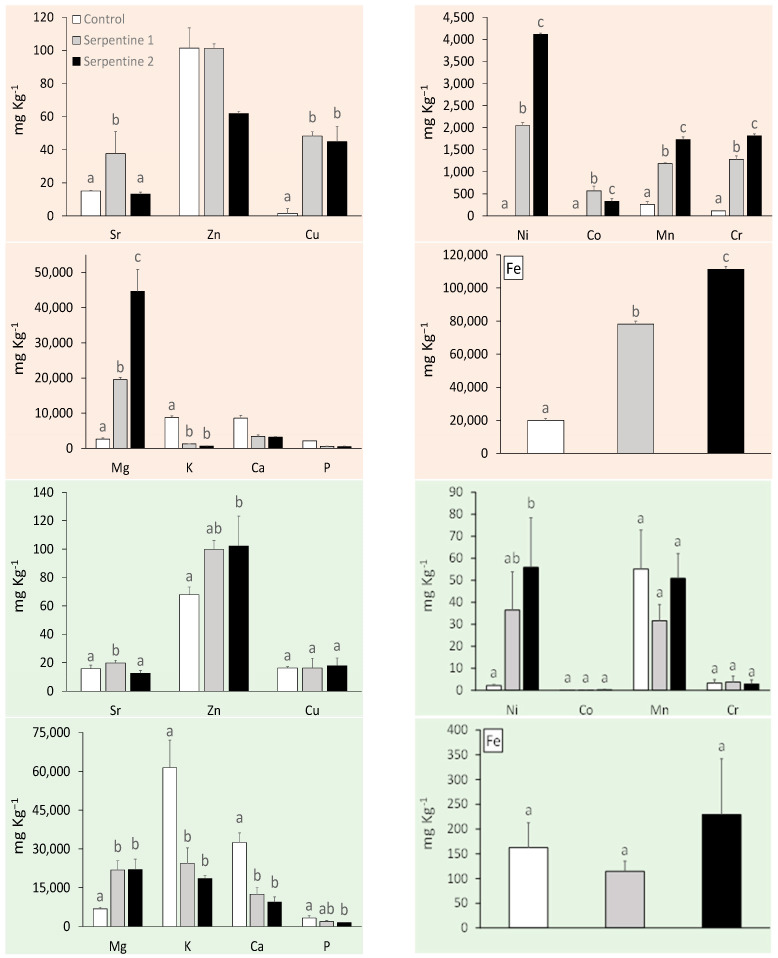
Trace metals and macronutrient concentrations (mean ± std) in soil (orange) and *B. radula* leaves (green) under the study treatments. Different letters indicate significant treatment differences (Tukey, *p* < 0.05).

**Figure 2 plants-11-02360-f002:**
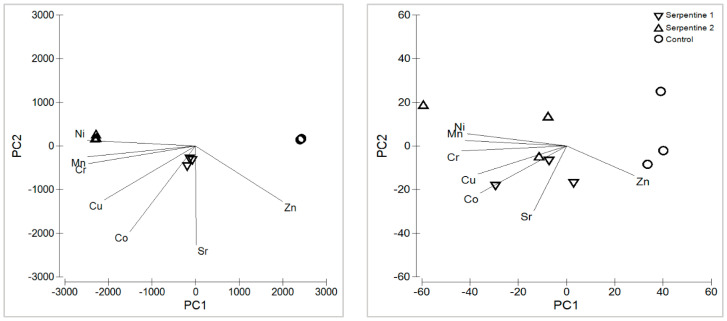
Principal component analysis of heavy metals in soil (**left**) and *B. radula* leaves (**right**), for the study treatments.

**Figure 3 plants-11-02360-f003:**
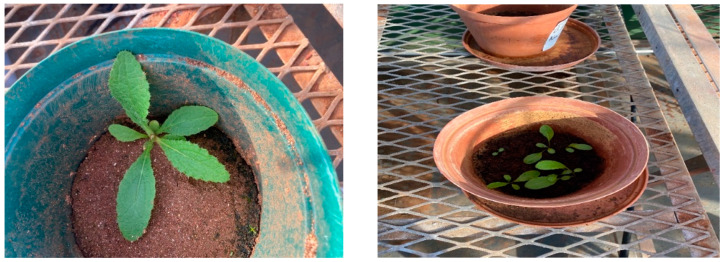
*B. radula* seedlings (90 d.a.p.) from Serpentine 2 (**left**) and control (**right**).

**Figure 4 plants-11-02360-f004:**
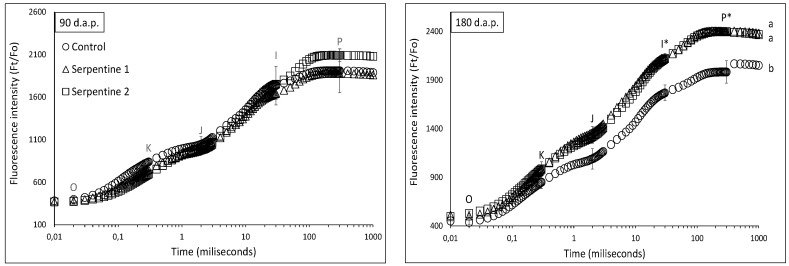
Chlorophyll *a* fluorescence transients (mean ± std) of *B. radula* leaves for the study treatments. Different lowercase letters indicate significant difference among treatment curves. * significantly different turnover steps among treatments (Tukey; *p* < 0.05).

**Figure 5 plants-11-02360-f005:**
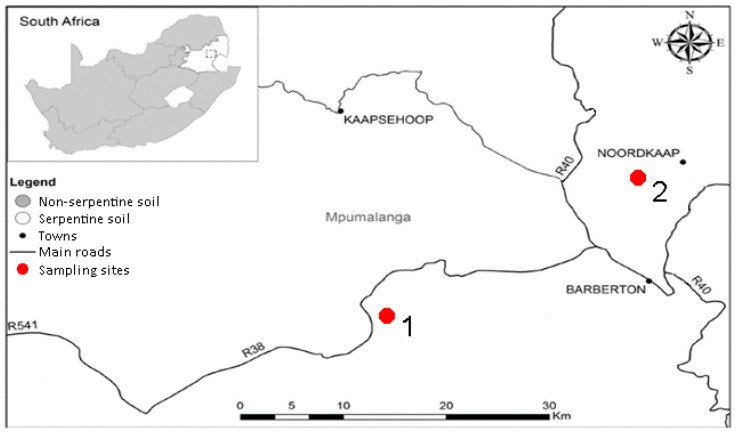
Sampling sites along the Barberton Greenstone Belt [16] (adapted with permission from Venter et al., Ecol. Res.; published by Springer Nature, 2017).

**Table 1 plants-11-02360-t001:** Principal component analysis results are based on the heavy metals for the soil treatments.

	Heavy Metal	PC1	PC2
Soils	Ni	−0.874	0.350
	Cr	−0.364	−0.469
	Mn	−0.313	−0.245
	Co	−0.075	−0.769
	Cu	−0.009	−0.044
	Zn	0.008	−0.041
	Sr	0.000	−0.047
Leaves	Ni	0.812	0.112
	Zn	0.570	0.046
	Mn	−0.121	0.982
	Cr	0.021	−0.006
	Cu	0.014	0.040
	Sr	−0.009	−0.138
	Co	0.003	0.002

**Table 2 plants-11-02360-t002:** Values (means ± std) of developmental and photosynthetic variables assessed in plants of *B. radula* under the study treatments. Different letters in the same column indicate significant differences among treatments.

Development										
	Germination (%)		Leaf Count		Leaf Length (mm)		Biomass (mg)			
D.a.p.	180		90	180	90	180	90	180		
Control	80.0 ± 8.2 a		3.7 ± 1.2 a	5.0 ± 1.0 a	53.7 ±15.9 a	78.6 ± 27.6 a	136.7 ± 49.3 a	503.3 ± 275.4 a		
Serpentine 1	62.5 ± 20.6 a		5.0 ± 1.0 a	6.0 ± 1.0 a	68.4 ± 13.1 a	108.4 ± 26.1 a	206.7 ± 75.7 a	1416.7 ± 877.8 a		
Serpentine 2	65.0 ± 12.9 a		5.0 ± 1.0 a	6.7 ± 1.2 a	39.6 ± 20.2 a	117.2 ± 23.1 a	165.0 ± 206.1 a	1750.0 ± 791.6 a		
Photosynthetic efficiency										
	ϕ_Po_/(1 − ϕ_Po_)		ψ_Eo_/(1 − ψ_Eo_)		δ_Ro_/(1 − δ_Ro_)		γ_RC_/((1 − γ_RC_))		PI_ABS,total_	
D.a.p.	90	180	90	180	90	180	90	180	90	180
Control	4.2 ± 0.4 a	3.9 ± 0.3 a	1.3 ± 02 a	1.2 ± 0.1 a	0.3 ± 0.0 a	1.2 ± 0.1 a	0.4 ± 0.0 a	0.4 ± 0.0 a	0.5 ± 0.1 a	0.6 ± 0.2
Serpentine 1	3.1 ± 1.8 a	4.0 ± 0.3 a	1.3 ± 0.4 a	1.3 ± 0.2 a	0.5 ± 0.3 a	1.3 ± 0.2 a	0.6 ± 0.2 a	0.4 ± 0.0 a	0.9 ± 0.3 ab	0.7 ± 0.1
Serpentine 2	4.2 ± 1.7 a	4.1 ± 0.2 a	1.5 ± 0.5 a	1.5 ± 0.2 a	0.5 ± 0.2 a	1.5 ± 0.6 a	0.5 ± 0.2 a	0.4 ± 0.0 a	1.6 ± 0.6 b	1.0 ± 0.3

**Table 3 plants-11-02360-t003:** Correlation coefficients among developmental parameters, photosynthetic efficiency and heavy metals and macronutrients in *B. radula* leaves, with significant relations among each other. GR = growth rate, LC = leaf count, LL = leaf length, BM = biomass. * Significant (*p* < 0.05).

	GR	LC	LL	BM	ψEo/(1 − ψEo)	RC/ABS	PI_TOTAL_	Zn	Ni	Co	K	Ca
Zn	−0.534	0.780	0.800 *	0.460	0.650	0.786 *	0.704 *	-	-	-	-	-
Ni	−0.510	0.830 *	0.660	0.580	0.664 *	0.930 *	0.842 *	0.884 *	-	-	-	-
Co	−0.428	0.780 *	0.780 *	0.380	0.688 *	0.854 *	0.847 *	0.903 *	0.863 *	-	-	-
K	0.778 *	−0.470	−0.560	−0.590	−0.670	−0.660	−0.565	−0.790 *	−0.770 *	−0.650	-	-
Ca	0.586	−0.590	−0.650	−0.680 *	−0.511	−0.825 *	−0.663	−0.790 *	−0.860 *	−0.670	0.899 *	-
P	0.560	−0.400	−0.350	−0.770 *	−0.500	−0.770	−0.598	−0.604	−0.738 *	−0.592	0.827 *	0.862 *

## Data Availability

Dataset containing concentrations of all evaluated elements in soil and plants are available at figshare.com. DOI: 10.6084/m9.figshare.21026137.

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
