# Peer review of "The Influence of Serpentine Soil on the Early Development of a Non-Serpentine African Thistle, Berkheya radula (Harv.) De Wild"

_plants, 2022, doi:10.3390/plants11182360_

Round 1

Reviewer 1 Report

In this study, authors planted a non-serpentine species, Berkheya radula, in three different types of soil, and investigated the plant growth, photosynthesis capacity, heavy metals in both plant and soil, to show its heavy metal tolerance. The paper is prepared carelessly, as many mistakes are showed in the manuscript. Authors should improve their manuscript.

1.       Line 44, they used “;” between the Ref number, but they used “,” in line 51.

2.       Line 72, there is a Ref cited as “35”, which is after Ref “14”.

3.       Line 92, “HNO3” should be “HNO3”.

4.       Line 131, the Ref 15, 16, 17 are missing.

5.       Line 161, the Ref 22 is missing.

6.       Line 230, the Ref 27 is missing.

7.       Line 229, “ZN” should be “Zn”.

8.       Line 230, I think there is no “,” after “their toxicity to plants”.

9.       There is no Ref 33, 34 cited in the text.

10.   The reference format should be adjusted according to this journal.

11.   Figure 3 and Table 2, all the significance should be showed. If there is no significance, the same letter can be used.

12.   Table 1 and 2, the “,” in the data is correct? It seems to be “.”.

13.   The tables should be adjusted to be more good-looking.

14.   The photos of the plant planted in the soil should be provided to support the data in Table 2.

15.   The abstract seems to be too long.

16.   Authors analyzed the trace elements in plant leaves, how about them in roots?

Author Response

Below is the attached file with the answers to the reviews.

Reviewer 2 Report

Two general comments:

1. Authors please review the document and be consistent -- verb tense- 3rd person,

2. Similarly -consistency either chemical name of Symbols.

Minor suggestions;

Lines 24-25 confusing- contradictory?

line 37 'problemtic'?  why? adapted plants grow well!

lines 56-62 why are we interested in growing non adapted plants?

line 67f Site 31, red soil?, site 2?

line 89 'equipment' of instrument ?

line 135 --- soils are derived---

Author Response

As reviewer number 2 your remarks are addressed in the attached document.

Reviewer 3 Report

The article in general is interesting but needs some in-depth analysis such as the insertion of images and further details that support the results.

I believe it is necessary:
  - insert images of the studied specie
- better argue the results obtained from a biochemical point of view.
Similar works already exist that report superficial and generic conclusions.

Author Response

Reviewer 3, the remarks have been addressed in the below document.

Round 2

Reviewer 1 Report

The manuscript is improved after the authors' revision.

Reviewer 3 Report

Accepted for publication